# 3′HS1 CTCF binding site in human β-globin locus regulates fetal hemoglobin expression

Pamela Himadewi[1†], Xue Qing David Wang[1†], Fan Feng[2†], Haley Gore[1], Yushuai Liu[1], Lei Yu[3], Ryo Kurita[4], Yukio Nakamura[5,6], Gerd P Pfeifer[1], Jie Liu[2]*, Xiaotian Zhang[1]*‡

[1]Center for Epigenetics, Van Andel Research Institute, Grand Rapids, United States; [2]Department of Computational Medicine and Bioinformatics, University of Michigan, Ann Arbor, United States; [3]Cell and Development Biology, University of Michigan, Ann Arbor, United States; [4]Department of Research and Development, Central Blood Institute, Japanese Red Cross Society, Tokyo, Japan; [5]Cell Engineering Division, RIKEN BioResource Research Center, Tsukuba, Japan; [6]Faculty of Medicine, University of Tsukuba, Tsukuba, Japan

**Abstract** Mutations in the adult β-globin gene can lead to a variety of hemoglobinopathies, including sickle cell disease and β-thalassemia. An increase in fetal hemoglobin expression throughout adulthood, a condition named hereditary persistence of fetal hemoglobin (HPFH), has been found to ameliorate hemoglobinopathies. Deletional HPFH occurs through the excision of a significant portion of the 3′ end of the β-globin locus, including a CTCF binding site termed 3′HS1. Here, we show that the deletion of this CTCF site alone induces fetal hemoglobin expression in both adult CD34+ hematopoietic stem and progenitor cells and HUDEP-2 erythroid progenitor cells. This induction is driven by the ectopic access of a previously postulated distal enhancer located in the *OR52A1* gene downstream of the locus, which can also be insulated by the inversion of the 3′HS1 CTCF site. This suggests that genetic editing of this binding site can have therapeutic implications to treat hemoglobinopathies.

**\*For correspondence:**
drjieliu@umich.edu (JL);
xiaozhan@umich.edu (XZ)

†These authors contributed equally to this work

**Present address:** ‡Department of Pathology, University of Michigan, Ann Arbor, United States

**Competing interest:** The authors declare that no competing interests exist.

## Introduction

The human β-globin locus consists of five globin genes embedded in the olfactory receptor cluster. During early development, these globin genes undergo gene switching from embryonic ε-globin (*HBE*) to fetal γ-globin (*HBG1/2*) and finally to adult β-globin (*HBB*). Inherited mutations in the *HBB* gene lead to dysfunction of the adult β-globin protein, causing hemoglobinopathies (*Bauer et al., 2012*). The symptoms of these disorders, including sickle cell disease and β-thalassemia, can be alleviated by persistent expression of fetal hemoglobin (hereditary persistence of fetal hemoglobin [HPFH]) throughout adulthood, which compensates for the mutant adult β-globin (*Bank, 2006*; *Hassell, 2010*). As such, multiple genome-editing strategies have been proposed to mimic HPFH as a treatment for hemoglobinopathies (*Bauer et al., 2012*; *Breda et al., 2016*; *Sankaran et al., 2008*; *Sankaran et al., 2011*; *Sankaran et al., 2009*; *Traxler et al., 2016*; *Xu et al., 2011*; *Ye et al., 2016*; *Zeng et al., 2020*). Two types of HPFH have been identified based on patient genetics. First is the non-deletional HPFH caused by point mutations in the BCL11A binding site at the *HBG1/2* promoters, and disruption of this transcriptional repressor binding leads to the activation of these genes (*Forget, 1998*; *Liu et al., 2018*; *Martyn et al., 2018*; *Traxler et al., 2016*). Second is the deletional HPFH that consists of the excision of a large genomic region within the β-globin locus, frequently including *HBB* and *HBD*

(*Forget, 1998*; *Ye et al., 2016*). These deletions can vary in length, and it remains unclear as to how they lead to the expression of fetal globin in adulthood (*Ye et al., 2016*).

## Results

The human β-globin gene locus is flanked by five CTCF binding sites (CBSs), which form the anchors for six chromosomal loops (*Huang et al., 2017*). Two convergent CBSs, designated as 3′HS1 and HS5, are located at the borders of the globin gene cluster. These two CBSs are nested between a downstream CBS (referred to as 3'-*OR52A5*-CBS) and two closely spaced upstream CBSs (referred together as 5'-*OR51B5*-CBSs). The HPFH deletions frequently cover the 3′HS1 CBS (*Figure 1A*). Therefore, we hypothesized that 3′HS1 may play a role in regulating β-globin cluster gene expression. To explore this, we first deleted the 3′HS1 using CRISPR/Cas9 genome-editing technology in K562 myelogenous leukemia cells, which express high levels of hemoglobin (*Figure 1—figure supplement 1A*). At the same time, we also deleted HS5 as a control in K562 cells. We observed that deletion of the HS5 CTCF site resulted in the upregulation of the 3′ genes including *HBB* and *HBG1/2*. Interestingly, the disruption of 3′HS1 CBS led solely to the upregulation of *HBG1/2* (*Figure 1—figure supplement 1B and C*, *Figure 1—figure supplement 1—source data 1*). These results show that altering the CTCF binding profile across the locus can significantly change the expression of the β-globin genes.

As HPFH deletions frequently cover the 3′HS1 CBS, we hypothesized that this site may contribute to the regulation of *HBG1/2*. To investigate further, we utilized the HUDEP-2 erythroid progenitor cell model, which predominantly express adult β-globin. We performed CRISPR/Cas9 editing to delete and invert the orientation of the 3′HS1 CBS to observe their respective impact on the expression of *HBG1/2* (*Figure 1B*). In bulk edited cells, we found that HS5 deletion did not alter globin gene expression significantly, yet 3′HS1 disruption did increase γ-globin gene expression as observed in K562 cells with low deletion percentage and low editing efficiency (*Figure 1—figure supplement 1D and E*). Subsequently, we generated two 3′HS1 CBS deletion clones (referred to as B6 and D3) and two 3′HS1 inversion clones (A2 and G3), whose genomic sequences were verified by Sanger sequencing and CTCF binding evaluated by CUT&RUN (*Figure 1D and E*, *Figure 1—figure supplement 1F*). To elucidate whether the genetic editing at 3′HS1 caused changes to gene expression, we differentiated the HUDEP-2 clones to activate β-globin expression.

We performed Hi-C and capture Hi-C to examine the changes to 3D chromatin organization at the β-globin locus following alterations to the CBSs (*Figure 1—figure supplement 2A*, *Figure 1C–E*). In situ Hi-C data was generated with high resolution at 5 kb. A total of 15,207–16,529 loops could be detected in the HUDEP-2 clones used for in situ Hi-C using Mustache (*Figure 1—figure supplement 2A*; *Roayaei Ardakany et al., 2020*). The CTCF bound around the β-globin locus form four chromosomal loops and separate the cluster into three distinct domains (*Figure 1A and C*, *Figure 1—figure supplement 2B*). Of notice, we could detect the enhancer to target gene interaction between the LCR and the *HBB* gene (*Figure 1—figure supplement 2B*). We also tested the copy number variance (CNV) in the three particular HUDEP-2 clones, we could verify all clones have chromosome number 49–50,XY, which is of normal range in unmodified HUDEP-2 cells (*Figure 1—figure supplement 2C*; *Moir-Meyer et al., 2018*; *Vinjamur and Bauer, 2018*). Next, we tested if the chromosomal loops were altered by the 3′HS1 editing. We applied the HiCCUPS method to call the significant chromosomal loops in the β-globin locus, four loops were identified with q value less than 0.1 (*Figure 1C and F*). We then use the q value of the called loops by HiCCUPS to quantify the strength of loop interactions between CBSs. Of the convergent CTCF interactions, 3′HS1 to 5'-*OR51B5*-CBSs was not called as loop with q value over 0.25. One loop was called between the two forward CTCF CBSs – 3′HS1 and 3′-OR52A5 CBS (*Figure 1D and F*). In the 3′HS1 deletion clone, the loss of CTCF at 3′HS1 resulted in the total loss of loops between 3′HS1 and HS5 as well as loops between 3′HS1 and 5'-*OR51B5*-CBSs (not called as loop). Concomitantly, a strong increase in the interaction between HS5 and 3'-*OR51A5*-CBS was observed (*Figure 1D and F*, *Figure 1—figure supplement 2A*). This reveals how the loss of a CTCF anchor drastically alters the 3D chromatin organization in the β-globin locus. The inversion of the 3′HS1 CTCF caused a significant increase in the interaction between 3′HS1 and 3'-*OR52A5 CBS*. Meanwhile, 3′HS1 upstream interactions with HS5 and 5'-*OR51B5*-CBSs were decreased (*Figure 1E and F*, *Figure 1—figure supplement 2A*). This revealed that the inversion of 3′HS1 CTCF drove the formation of chromosomal loops between the convergent CBSs, which may lead to stronger insulation of regulatory elements.

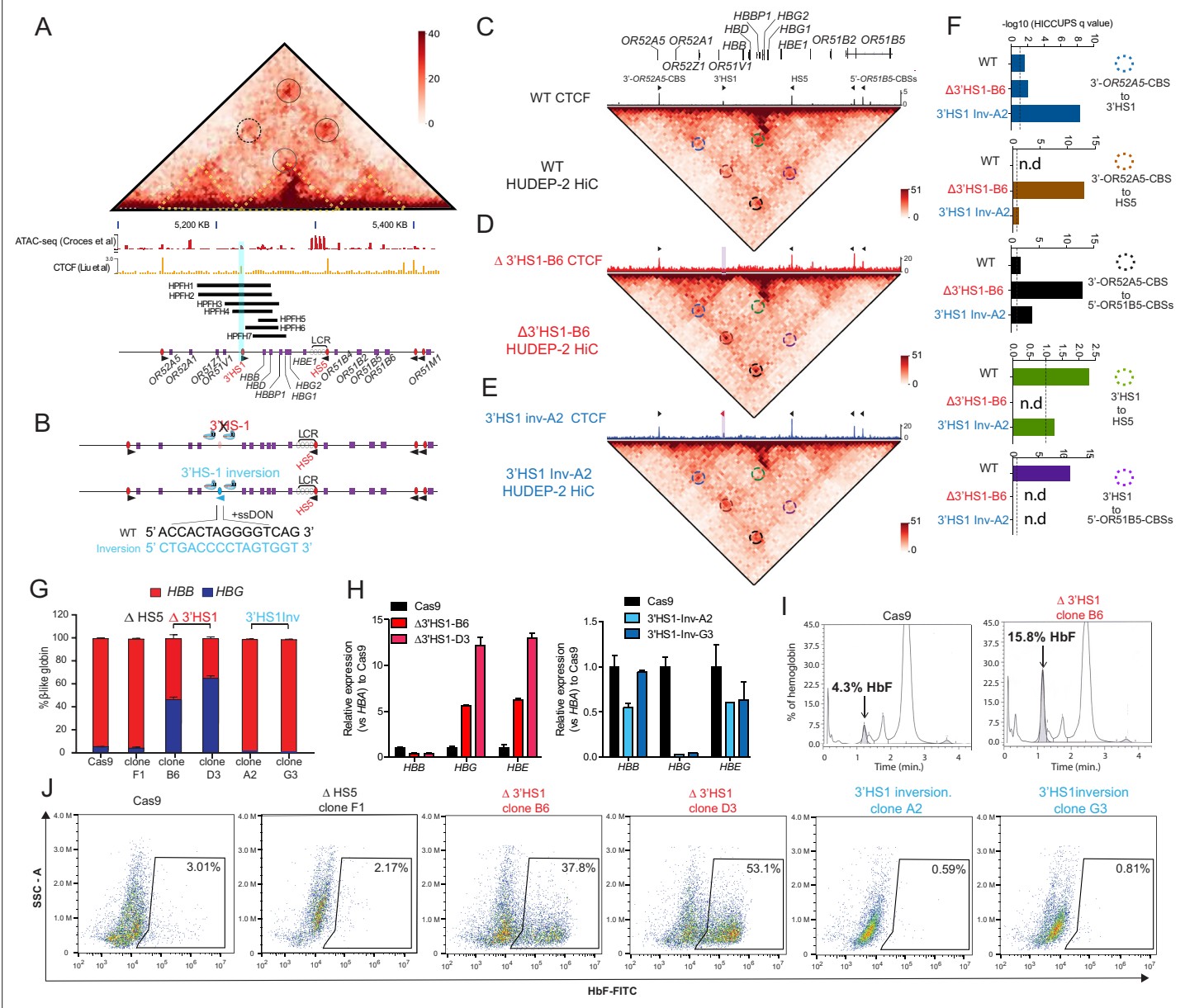

**Figure 1.** 3'HS1 modulates the hemoglobin gene expression in β-globin gene cluster. (**A**) Genome-wide Hi-C interaction map and regulatory landscape around β-globin gene cluster in human HUDEP2 cells. ATAC-seq and CTCF track of HUDEP2 cells (*Liu et al., 2018*) is shown in the lower panel. Black cycle indicates the position of loops previously identified (*Huang et al., 2017*). Yellow dotted line indicates the three sub-TAD domains identified previously (*Huang et al., 2017*). HPFH1-7 deletion is illustrated and 3'HS1 is marked in blue shade. (**B**). The scheme of CTCF binding motif orientation engineering in HUDEP-2 cells. (**C–E**) In situ Hi-C contact map around β-globin gene cluster in HUDEP-2 cells of wild type (**C**), 3'HS1 deletion (**D**), and 3'HS1 inversion (**E**). CTCF CUT&RUN tracks of WT (*Liu et al., 2018*), 3'HS1 deletion and 3'HS1 inversion HUDEP-2 cells are shown on the top of corresponding Hi-C plots. All loops that called in the HUDEP2 cells of three genotypes are marked with circles of different colors. (**F**) The HiCCUPS quantification of loops interaction strength by q value in β-globin locus. Dotted line annotates q = 0.1. n.d.: not detected by HiCCUPS (q value > 0.1). (**G**) The composition of β-like globin HUDEP-2 cells with 3'HS1 deletion. qPCR measurement of β-like globin HUDEP-2 in two clones (B6 and D3) of Δ3'HS1 HUDEP-2 cells is shown. Mean ± SD is displayed, n = 3. (**H**) Left panel: relative expression of *HBE, HBG* (probe measures both *HBG1* and *HBG2*), and *HBB* in the 3'HS1 deleted HUDEP-2 clone B6. Mean ± SD is displayed, n = 3. Right panel: relative expression of *HBE, HBG* (probe measures both *HBG1* and *HBG2*), and *HBB* in the 3'HS1 inverted HUDEP-2 clone A2. Mean ± SD is displayed, n = 3. (**I**) The right panel shows the High-performance liquid chromatography (HPLC) for globin composition in Cas9-treated HUDEP-2 control and 3'HS1 deletion clone B6. (**J**) Flow cytometry plot of HbF in HUDEP-2 cell clones with 3'HS1 deletion (B6 and D3), 3'HS1 inversion (A2 and G3), and ΔHS5 clone.

The online version of this article includes the following figure supplement(s) for figure 1:

**Figure supplement 1.** CTCF binding site around β-globin gene cluster regulated β-globin gene expression.

*Figure 1 continued on next page*

*Figure 1 continued*

**Figure supplement 1—source data 1.** The gel picture of paired guide deletion for HS5 and 3'HS1.

**Figure supplement 2.** 3D genomics change in Δ3'HS-1 clones and 3'HS-1 inversion HUDEP-2 cell clones.

**Figure supplement 3.** (**A**) The differentiation stage of HUDEP-2 cell clones used in *Figure 1*. (**B**) Immunofluorescence staining of HbF (top panel) and HBB (bottom panel) from clones used in *Figure 1*.

Next, we evaluated the expression of the β-globin genes and found that the *HBG1/2* and *HBE* genes upregulated 2.5- to 8-fold in the Δ3'HS1 clones (*Figure 1G and H*). In contrast, the inversion of 3'HS1 resulted in a >50% reduction of *HBE* and near-complete depletion of *HBG1/2* (*Figure 1H*). Most notably, the increase in *HBG1/2* upon deletion of the 3'HS1 CTCF site leads to a significant increase in fetal hemoglobin HbF (*Figure 1I*). Consequently, we evaluated the clones for HbF+ cells by flow cytometry. Consistent with transcription level, we observed an increase in HbF+ cells from 4.3 % in the Cas9 control clone to 37.8% and 53.1% in Δ3'HS1 B6 and D3 clones, respectively (*Figure 1J*). Meanwhile, inversion of 3'HS1 resulted in a decrease of HbF+ cells to below 1% . In contrast, deletion of the upstream HS5 CBS did not induce nor abrogate the quantity of HbF+ cells (*Figure 1J*). All these clones are well differentiated at the same stage when HbF is measured by flow cytometry (*Figure 1— figure supplement 3*).

We then performed ATAC-seq and H3K27ac ChIP-seq in the Δ3'HS1 and 3'HS1 inversion clones to examine the regulatory landscape of the β-globin gene cluster. Following 3'HS1 CBS deletion, we observed significant open chromatin at the *HBG1/2*, *BGLT3,* and *HBBP1* genes (*Figure 2—figure supplement 1A*). There was also a significant increase in activating H3K27ac in the *HBG2* gene body. Interestingly, these epigenetic changes upon deletion of 3'HS1 CBS do not occur at the promoter of *HBG2,* which suggests that this 3'HS1-dependent regulation is independent from BCL11A transcriptional regulation (*Figure 2—figure supplement 1A*).

To determine if other regulatory pathways, such as the transcriptional repressor BCL11A, are involved in the upregulation of γ-globin expression in Δ3'HS1 and 3'HS1 inversion clones, we performed RNA-seq to identify differentially expressed genes in these clones. We found 161 upregulated and 153 downregulated genes in the Δ3'HS1 clones with *HBG1/2* genes being the most significantly upregulated as well as β-globin cluster genes *HBE1*, *HBBP*, and *BGLT3* (*Figure 2A and C*). In the 3'HS1 inversion clones, we identified only 3 upregulated genes and 51 downregulated genes (*Figure 2B*). In these clones, we observed downregulation of *HBG2* and upregulation of the nearby *OR52A5* gene; however, we did not observe any change in the expression of known γ-globin regulators (*BCL11A, ZBTB7A (LRF), ELF2AK1 (HRI), ATF4, ZNF410*, and *NFIX*) (*Figure 2B and D*, *Supplementary file 1*; *Grevet et al., 2018*; *Masuda et al., 2016*). We further verified BCL11A protein level by immunoblotting and observed no significant reduction in Δ3'HS1 clones (*Figure 2*, *Figure 2—source data 1*). Taken together, we concluded that BCL11A does not contribute to the induction of HbF in the edited HUDEP-2 cells. To further elucidate the role of BCL11A in the regulation of *HBG1/2*, we performed targeted disruption of a known *BCL11A* gene enhancer in both WT and Δ3'HS1 HUDEP-2 clones (*Figure 2—figure supplement 1B*, *Figure 2—figure supplement 1—source data 1*; *Bauer et al., 2013*). This mutation resulted in a significantly lower BCL11A expression, and we observed further increase of *HBG1/2* expression level and HbF+ cells (*Figure 2F*, *Figure 2—figure supplement 1C and F*). Pomalidomide was found to boost the level of HbF in adult erythroblasts by destabilizing the BCL11A protein in cells (*Grevet et al., 2018*). Therefore, we treated HUDEP-2 3'HS1 deletion clones with pomalidomide and observed a significant reduction in BCL11A protein. This led to further increase in expression of fetal globin (*Figure 2—figure supplement 2A and D*, *Figure 2—figure supplement 2—source data 1*). These results show that genetic editing of the regulatory *cis*-element (3'HS1 CBS) accentuates with depletion of the transcriptional repressor BCL11A. This further indicates that the γ-globin activation in 3'HS1 deletion HUDEP-2 cells was not driven by the BCL11A-associated pathways. The double disruption of BCL11A and 3'HS1 also led to similar level of γ-globin activation as the disruption of BCL11A alone (*Figure 2F*). This data suggests that 3'HS1-regulated γ-globin repression might be hypostatic to the BCL11A-mediated γ-globin repression.

Deletional HPFH has been proposed to be the result of distal enhancer juxtaposition in the region downstream of 3'HS1, and several HPFH enhancers have been identified (*Forget, 1998*). The reduction of HbF in the 3'HS1 inversion HUDEP-2 clones suggests that potential enhancers may be located between 3'HS1 and 3'-*OR52A5*-CBS. These enhancers may be insulated by the loop formed between

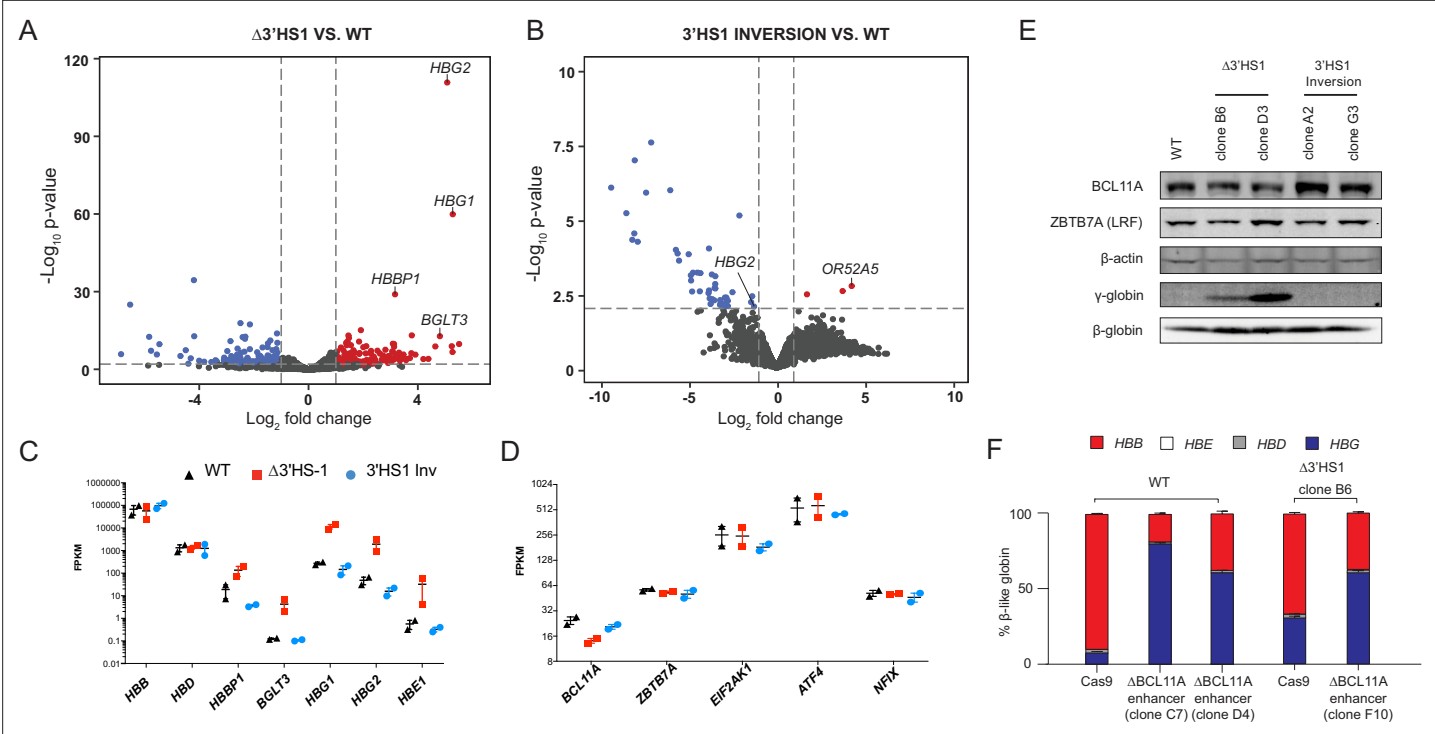

**Figure 2.** The induction of HbF in Δ3'HS1 cells is BCL11A independent. (**A**) Volcano plot of differentially expressed genes in two Δ3'HS1 clones (B6 and D3) vs. two wild-type HUDEP-2 biological duplicates. Differentially expressed globin and olfactory receptor genes are labeled. (**B**) The volcano plot of differentially expressed genes in two 3'HS1 inversion clones (A2 and G3) vs. two wild-type HUDEP-2 biological duplicates. Differentially expressed globin and olfactory receptor genes are labeled. (**C**) Expression level of β-globin genes in Δ3'HS1 clones, 3'HS1 inversion clones, and wild-type HUDEP-2 cells. (**D**) Expression level of known fetal hemoglobin repressor genes in Δ3'HS1 clones, 3'HS1 inversion clones, and wild-type HUDEP-2 cells. (**E**) Western blot shows the level of BCL11A and ZBTB7A (LRF) in Δ3'HS1 clones, 3'HS1 inversion clones, Δ3'HS-5 clones, and wild-type HUDEP-2 cells. Refer to *Figure 2—source data 1* for original blot picture. (**F**) The composition of β-like hemoglobin genes in the WT HDUEP-2 cells with *BCL11A* + 58 enhancer deleted with CRISPR/Cas9 and Δ3'HS1 HDUEP-2 cells with *BCL11A* + 58 enhancer deleted with CRISPR/Cas9.

The online version of this article includes the following source data and figure supplement(s) for figure 2:

**Source data 1.** The immunoblot data of BCL11A, ZBTB7A, β-actin, β-globin, and γ-globin of clones displayed in *Figure 2*.

**Figure supplement 1.** BCL11A loss further promotes fetal hemoglobin induction in Δ3'HS-1 background.

**Figure supplement 1—source data 1.** The immunoblot data of BCL11A, ZBTB7A, β-actin, β-globin, and γ-globin of clones displayed in *Figure 2—figure supplement 1*.

**Figure supplement 2.** Pomalidomide enhances fetal hemoglobin production induced by 3'HS-1 deletion.

**Figure supplement 2—source data 1.** The immunoblot data of BCL11A, ZBTB7A, β-actin, β-globin, and γ-globin of clones displayed in *Figure 2—figure supplement 2*.

3'HS1 to 3'-*OR52A5*-CBS in the inversion clones. With no significant change in chromatin interaction observed between the LCR and *HBG1/2* genes (*Figure 1—figure supplement 2C and D*), it suggests that the *HBG1/2* expression increase is controlled by a *cis*-element other than the LCR. We then analyzed the ATAC-seq, GATA1 ChIP-seq data together to search for potential *cis*-regulatory elements. We found the *OR52A1* region bound by the GATA1 transcriptional activator in the erythroid lineages (*Corces et al., 2016*; Feingold and Forget; *Liu et al., 2018*; *Figure 3A*). Importantly, the mapping of a previously described HPFH enhancer encompassed both *OR52A1* gene and GATA1 binding site (*Elder et al., 1990*; *Feingold and Forget, 1989*). We proceeded to delete the GATA1 binding site within the HPFH enhancer site and the entire HPFH enhancer marked by ATAC-seq in Δ3'HS1 clones (*Figure 3B*). We found that both deletions reduced the *HBG1/2* expression and HbF+ cell percentage in these cells, suggesting that the HPFH enhancer contributes to the activation of *HBG1/2* (*Figure 3C–E*). We further evaluated whether there were direct interactions between *HBG1/2* and HPFH enhancer after the deletion of 3'HS1 by virtual 4C (v4C) in our Hi-C dataset. We observed mildly increased v4C signal in *HBG2* promoter region of Δ3'HS1 cells compared with WT and 3'HS1 inversion clones when HPFH

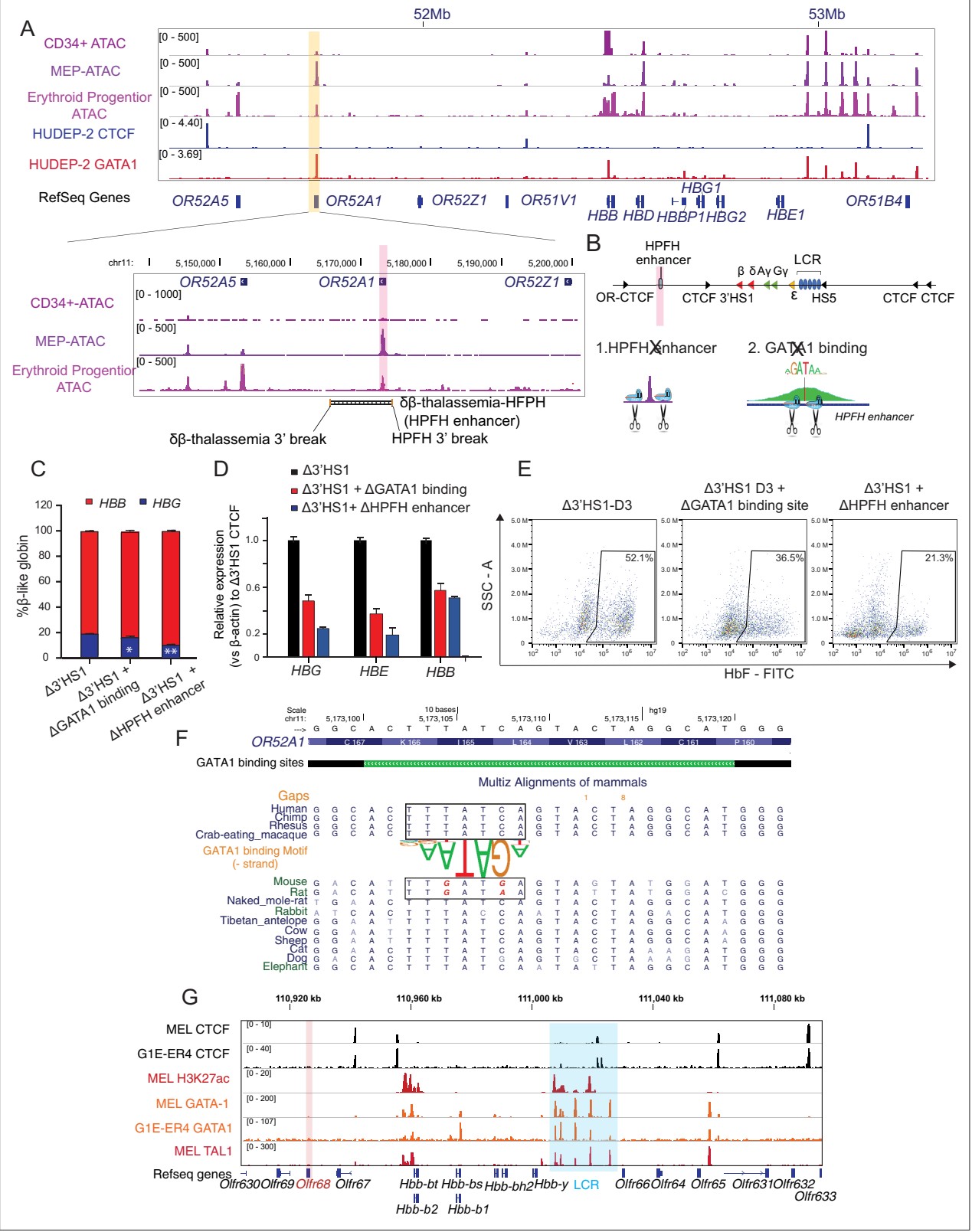

**Figure 3.** The induction of HbF in Δ3'HS1 cells is modulated by HPFH enhancer. (**A**) Upper panel: IGV view of ATAC-seq in primary human blood cells followed by GATA1 and CTCF ChIP-seq in HUDEP-2 cells around the β-globin locus. Lower panel: ATAC-seq of CD34+ hematopoietic stem and progenitor cell (HSPC), megakaryocyte–erythroid progenitor (MEP), and erythroblast is shown in the zoomed view for the *OR52A1* region. Red-shaded area indicates the locus of *OR52A1*. HPFH 3' beak and δβ-thalassemia 3' break is annotated (***Feingold and Forget, 1989***). (**B**) The experimental scheme

*Figure 3 continued on next page*

*Figure 3 continued*

of hereditary persistence of fetal hemoglobin (HPFH) deletion in the 3'HS1 deletion background. (**C**) The composition of β-like globin Δ3'HS1 (clone B6) HUDEP-2 cells with GATA1 binding site and HPFH region deletion. Mean ± SD is displayed, n = 3. (**D**) Relative expression of *HBE, HBG* (probe measures both *HBG1* and *HBG2*), and *HBB* in the Δ3'HS1 (clone B6) HUDEP-2 cells with GATA1 binding site and HPFH enhancer region deletion. Mean ± SD is displayed, n = 3. (**E**) The representative HbF flow plot of Δ3'HS1 (clone B6) HUDEP-2 cells with GATA1 binding site and HPFH enhancer region deletion. (**F**) Evolution conservation of OR52A1 GATA1 binding site in vertebrates. GATA1 binding motif is shown in the middle. The site in mouse and rat associated with human GATA1 binding is boxed out. (**G**) Chromatin landscape of mouse β-globin gene cluster in mouse erythroid cells MEL and G1-ER4. CTCF, GATA1, and TAL1 ChIP-seq is shown. Orange stripe highlights the mouse homolog of human *OR52A1–Olfr68*.

The online version of this article includes the following figure supplement(s) for figure 3:

**Figure supplement 1.** HPFH enhancer in edited HUDEP-2 cells.

enhancer region acts as viewpoint (***Figure 3—figure supplement 1A***). Vice versa, we also observed mildly increased v4C signal in HPFH enhancer region of Δ3'HS1 cells compared with WT and 3'HS1 inversion clones when *HBG2* promoter region acts as viewpoint (***Figure 3—figure supplement 1A***). These data suggest that the deletion of 3'HS1 might have induced more frequent interaction between HPFH enhancer and *HBG2* promoter regions.

Previously, mouse and human β-globin locus and its surrounding regions have been shown to be evolutionarily conserved (***Bulger et al., 2003***). More interestingly, previous report showed that the disruption of 3'HS1 site in mouse did not result in any change in the β-like globin gene expression (***Bender et al., 2006***). We hypothesized that the alteration of HPFH enhancer sequence might contribute to the different outcome of 3'HS1 deletion in mouse and human. Therefore, we evaluated the evolutionary conservation of *OR52A1* in mammals and found mouse and rat homolog of human *OR52A1–Olfr68* bears two single-nucleotide substitutions right at the core binding site of GATA1 (***Figure 3F***). When we further checked the GATA1 ChIP-seq in mouse erythroid cells, we also found the absence of binding in the *OR52A1*'s homolog *Olfr68* and its surrounding region (***Figure 3G***). This data suggests that the mouse olfactory receptor region 3' to β-globin genes no longer bears GATA1 binding sites and enhancer activity. Overall, the loss of GATA1 binding in mouse clearly explains the difference between mouse and human on the effect of 3'HS1 deletion on globin gene expression.

Previously, it has been proposed that the juxtaposition of HPFH enhancer results in the activation of γ-globin. We wonder if the chromosomal distance between HPFH enhancer and globin genes also contributes to the regulation of globin gene expression. We used paired guide RNA to delete a 48 kb region between HPFH enhancer and 3'HS1, to access the effect of distance between enhancer and target genes in the presence of chromosomal insulators (***Figure 3—figure supplement 1B***). We obtained a heterozygous clone bearing this 48 kb deletion (***Figure 3—figure supplement 1B***). We found that moving the HPFH enhancer to the proximity of globin locus mildly increases the *HBG1/2* gene expression by threefold with increase of *HBE1* gene and reduction of *HBB* gene (***Figure 3—figure supplement 1C and D***). Overall, the data suggests that both chromosomal distance and insulator contribute to the low expression of *HBG1/2*, but chromosomal insulators are dominant to insulate HPFH enhancer to access the globin genes.

To assess the therapeutic potential of 3'HS1 deletion in primary HSPCs, we performed the 3'HS1 and HS5 CBS deletions in adult mobilized peripheral blood CD34+ HSPC from three different donors (***Figure 4A***). We achieved high deletion percentage of both CBSs in these primary cells (***Figure 4B***, ***Figure 4—source data 1***). Upon differentiation, we observed a robust increase in HbF+ cells across all three donors with 3'HS1 deletion but not with HS5 deletion (***Figure 4C and D***). The normal erythroid differentiation was not affected by either CBS deletion (***Figure 4—figure supplement 1A and B***). While the disruption of the 3'HS1 CBS in primary patient cells did not yield as great of an effect on HbF+ cells as in HUDEP-2 cells, the results do support the involvement of a downstream *cis*-acting regulatory element on the *HBG1/2* genes.

Finally, based on our data, we propose a model where the 3'HS1 CBS modulates the HPFH enhancer's access to the *HBG1/2* genes (***Figure 4E***).

## Discussion

Here, we show that the deletion of a CBS – 3'HS1– in the human β-globin locus can phenocopy HPFH. This condition is driven by alteration of the 3D genomic organization around the β-globin

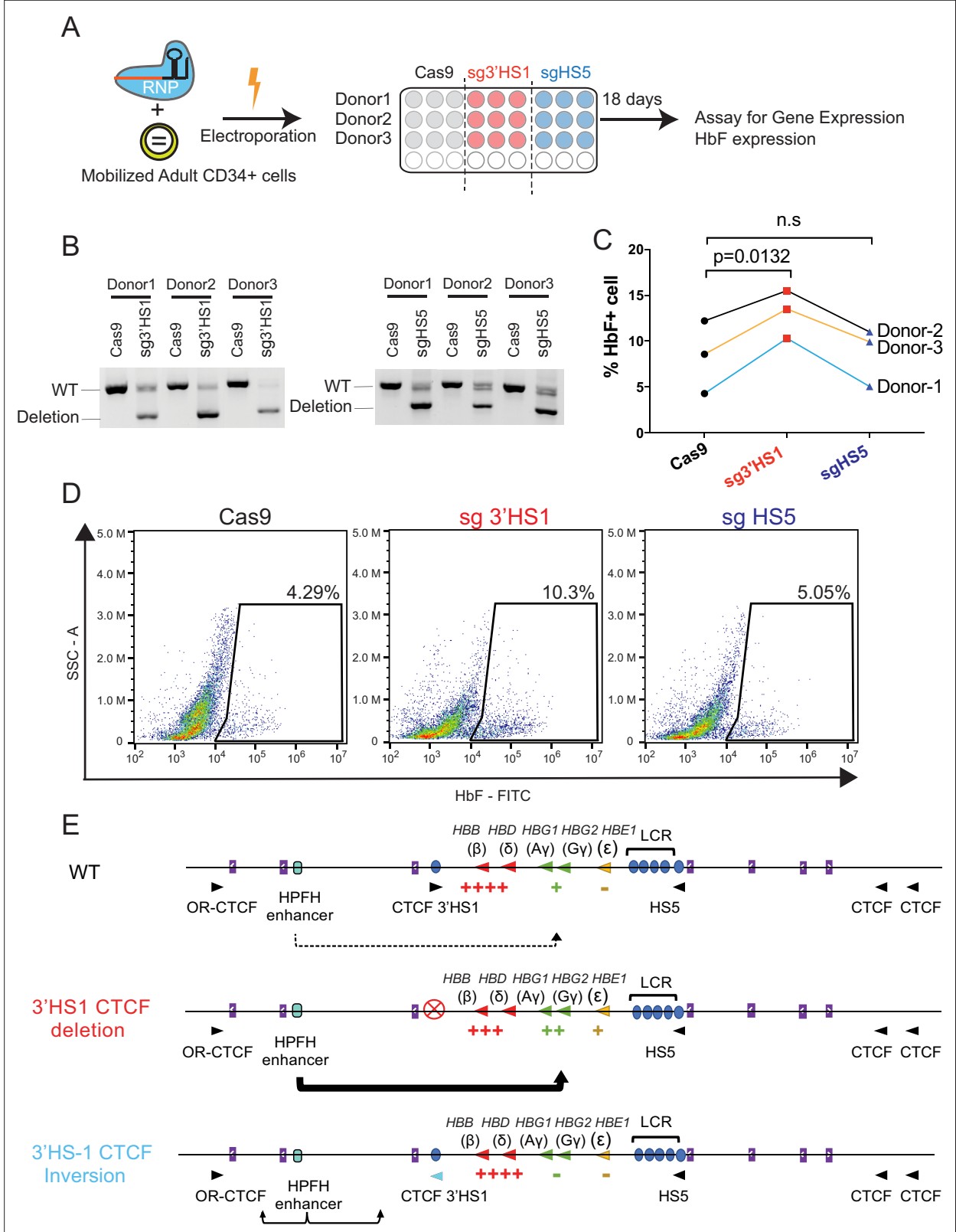

**Figure 4.** Deletion of 3'HS1 induces HbF in primary adult HSPC. (**A**) The experimental scheme for primary HSPC editing. (**B**) The deletion of 3'HS1 and HS5 in three CD34+ peripheral blood mononuclear cell (PBMC) HSPCs from three individual adult donors. Refer to *Figure 4—source data 1* for original gel picture. (**C**) The HbF+ cell percentage at day 21 in three HSPCs from three individual adult donors after 3'HS1 and HS5 deletion. p-Value is calculated by one-tailed paired t-test. n.s., not significant, p=0.3659 in HS5 deletion vs. Cas9 by one-tailed paired t-test. (**D**) The reprehensive flow

*Figure 4 continued on next page*

*Figure 4 continued*

plot for HbF+ cells at day 21 in 3'HS1-deleted and HS5-deleted PBMC HSPC. The data is from donor #1. (**E**) The model of fetal hemoglobin regulation through 3'HS1.

The online version of this article includes the following figure supplement(s) for figure 4:

**Source data 1.** The gel picture of paired guide deletion for HS5 and 3'HS1 in HSPC.

**Figure supplement 1.** 3'HS1 deletion in HSPC.

locus, which allows the long-range interaction of a distal enhancer in the *OR52A1* gene to drive the expression of *HBG1/2*. We further show that inversion of 3'HS1 insulates the enhancer element and further suppresses *HBG1/2*. Previously, induced LCR to *HBG1/2* interaction showed the importance of high-order chromatin structure in the regulation of globin gene expression (***Breda et al., 2016***; ***Deng et al., 2014***). However, the role of CTCF protein bound around the gene cluster was not clear. Our study reveals how CTCF binding at this locus modulates the accessibility of the fetal *HBG1/2* genes to a downstream enhancer. In the HPFH enhancer scenario, 3'HS1 limits the HPFH enhancer access to *HBG1/2* by forming the sub-TAD with 5'HS (***Figure 4E***; ***Oudelaar et al., 2021***). When the 3'HS1 CBS is deleted, the HPFH enhancer gains access to *HBG1/2* without the hinder of 3'HS1–HS5 loop. When the 3'HS1 CBS motif is inverted, the HPFH enhancer is further restricted by the pairing of 3'-*OR52A5*-CBS to the inverted 3'HS1 CBS, which results in the strong chromosomal loop formation between the two CBSs. This insulation leads to the reduced *HBG1/2* expression and upregulation of *OR52A5* (Figures 2B and 4E).

Furthermore, we have elucidated the function of long speculated HPFH enhancer in the induction of HbF with our Δ3'HS1 cells. However, we suspect that other regulatory elements also exist between 3'HS1 and 3'-*OR52A5*-CBS since we only get a partial decrease of *HBG1/2* expression with HPFH enhancer deletion in Δ3'HS1 cells. This may suggest that other previously identified HPFH enhancers also contribute to HbF induction in the Δ3'HS1 cells (***Feingold and Forget, 1989***; ***Forget, 1998***). Indeed, other GATA1 binding sites can be observed between *OR52A1* and *OR52A5* genes of erythroid progenitor cells. Overall, these observations suggest that GATA1 binding across the locus works collectively to activate *HBG1/2* expression with or without direct promoter-enhancer interactions. Interestingly, despite the evolution conservation of *OR52A1* gene in mammals, TF binding disrupting nucleotide substitutions occur at the GATA1 binding site in HPFH enhancer in some mammal species that do not express distinct form of fetal hemoglobin (mouse and rat) (***Figure 3F and G***). This evolutionary alteration in the TF binding site suggests that HPFH enhancer may also play a role in regulating globin expression in other developmental stages. The consistent binding of GATA1 in high *HBE/HBG*-expressing K562 cells and high *HBB*-expressing HUDEP-2 cells suggests that the potential function of HPFH enhancer may still need the modulation of 3D genomic reorganization.

Despite our data suggesting that deletion of 3'HS1 is sufficient to induce the γ-globin activation, yet, the upregulation does not completely phenocopy the HPFH condition, in which the HbF is expressed at pancellular level. The deletion of 3'HS1 only induces a portion of cells to be *F+* cells, with an interesting differentiation block phenotype in a large portion of clones we have selected (***Figure 1J***). Although HUDEP-2 is known to be heterogenous in clonal level, the *F+* cell phenotype of 3'HS1-deleted cells suggests that the full HPFH phenotype may require the deletion of *HBB* and *HBD* genes to erase the strong enhancer-promoter interaction between LCR and *HBB*. Furthermore, our results indicate that the distance between HPFH enhancer and the globin locus could also play a role in regulating γ-globin gene expression. This data hints that the deletion of both 3'HS1 and the flanking region between 3'HS1 and HPFH enhancer may activate the fetal hemoglobin to an even higher level. Larger deletions will result in the reorganization of chromosomal loop interactions as well as the decrease of physical distance from HPFH enhancer to globin locus. The enhancers have to locate at certain range of distance from the target gene promoters to exert the maximal activation effect (***Jessica Zuin et al., 2021***). Therefore, a method that can create large deletion that disrupts 3'HS1 and reduces enhancer-promoter distance simultaneously could be a potential gene therapy to effectively activate the γ-globin expression in hemoglobinopathies.

The differentiation stage of HUDEP-2 cell clones used in ***Figure 1*** was profiled by flow cytometry of CD71 and CD235a (***Figure 2—figure supplement 1—source data 1***). The immunoblot data of

BCL11A, ZBTB7A, β-actin, β-globin, and γ-globin of clones is displayed in *Figure 2—figure supplement 2*.

# Materials and methods

## Key resources table

| Reagent type (species) or resource | Designation | Source or reference | Identifier | Additional Information |
|---|---|---|---|---|
| Antibody | CTCF (rabbit polyclonal) | Abcam | AB70303 | WB(1:1000) |
| Antibody | BCL11A (rabbit polyclonal) | Abcam | AB191401 | WB(1:1000) |
| Antibody | β-Actin (rabbit polyclonal) | Proteintech | 20536-1-AP | WB(1:2000) |
| Antibody | β-Globin (mouse monoclonal) | Santa Cruz Biotechnology | SC-21757 | WB(1:2000) |
| Antibody | γ-Globin (mouse monoclonal) | Santa Cruz Biotechnology | SC-21756 | WB(1:500) |
| Antibody | ZBTB7A (mouse monoclonal) | R&D systems | MAB3496 | WB(1:1000) |
| Antibody | Human HbF-FITC (recombinant) | Miltenyl Biotec | 130-108-241 | FC(1:100) |
| Antibody | Human CD71-PE (mouse monoclonal) | BioLegend | 334105 | FC(1:100) |
| Antibody | Human CD235a-APC (mouse monoclonal) | BD Biosciences | 561775 | FC(1:100) |
| Antibody | Starbright B700-conjugated goat α-rabbit IgG (goat polyclonal) | Bio-Rad | 12004161 | WB(1:2000-1:5000) |
| Antibody | DyLight 800 goat α-mouse IgG (goat polyclonal) | Bio-Rad | STAR117D800GA | WB(1:2000-1:10,000) |
| Antibody | Acetyl-histone H3 (Lys27) (rabbit polyclonal) | Cell Signaling | 8173S | 2 µg per ChIP |
| Peptide, recombinant protein | SCF | Peprotech | 300-07 | |
| Peptide, recombinant protein | FLT3L | Peprotech | 300-19 | |
| Peptide, recombinant protein | TPO | Peprotech | 300-18 | |
| Peptide, recombinant protein | EPO | Amgen | EPOGEN | |
| Peptide, recombinant protein | IL-3 | Peprotech | 200-03 | |
| Other | SFEM II | STEMCELL Technologies | 09655 | |
| Chemical compound, drug | Dexamethasone | Sigma | D2915 | |
| Chemical compound, drug | Doxycycline | Sigma | D9891 | |
| Peptide, recombinant protein | Recombinant human insulin | Sigma | I2643 | |

*Continued on next page*

*Continued*

| Peptide, recombinant protein | Holo-transferrin | Sigma | T4132 | |
|---|---|---|---|---|
| Chemical compound, drug | Heparin | Sigma | H3393 | |
| Other | Human AB serum | Sigma | H6914 | |
| Peptide, recombinant protein | Cas9 Protein | IDT | 1081058 | |
| Commercial assay or kit | Concanavalin A Beads | Bangs Laboratories, Inc | BP531 | |
| Peptide, recombinant protein | pA-MNase | Gift from Steven Henikoff | | |
| Commercial assay or kit | Dynabeads protein A | Thermo Fisher Scientific | 1002D | |
| Commercial assay or kit | Dynabeads MyOne Streptavidin T1 | Thermo Fisher Scientific | 65601 | |
| Chemical compound, drug | Protease Inhibitor Cocktail | GenDEPOT | 50-101-5486 | |
| Cell line (*Homo sapiens*) | HUDEP-2 cells | Riken Cell Bank | RCB4557 | |
| Cell line (*Homo sapiens*) | K562 cells | ATCC | CCL-243 | |
| Biological sample (primary cells *Homo sapiens*) | Human peripheral blood CD34+ HSPCs | STEMCELL Technologies | 70060.1 | CD34+ HSPC isolated from individual donor. Sex is mixed. |
| Commercial assay or kit | Rapid RNA library kit | Swift Biosciences | R2096 | |
| Commercial assay or kit | Nextera XT library preparation kit | Illumina | FC-131-1024 | |
| Commercial assay or kit | MinElute PCR purification kit | Qiagen | 28004 | |
| Commercial assay or kit | Accel-NGS 2 S Plus DNA library kit | Swift Biosciences | 21096 | |
| Commercial assay or kit | 2 S Combinatorial Dual Indexing Kit | Swift Biosciences | 28096 | |
| Commercial assay or kit | HiC Next Generation Sequencing Kit | Arima Genomics | | |
| Commercial assay or kit | KAPA library quantification kit | KAPA Biosystems | KK4844 | |
| Commercial assay or kit | RNA clean & concentrator | Zymo Research | R1013 | |
| Other | Raw and processed NGS sequencing data | This paper | GSE160425 | Raw and processed data could be obtained from the link : https://www.ncbi.nlm.nih.gov/geo/query/acc.cgi?acc=GSE160425 |
| Other | HUDEP-2 GATA1 CUT&RUN | GEO: GSE104676 | GSM2805376 | |
| Other | HUDEP-2 CTCF ChIP-seq | GEO: GSE104676 | GSM3671075 | |

*Continued on next page*

| Other | HUDEP-2 BCL11A ChIP-seq | GEO: GSE103445 | GSM2771529 | |
|---|---|---|---|---|

*Continued*

| | | | | |
|---|---|---|---|---|
| Other | Hematopoietic cells differentiation ATAC-seq | *Corces et al., 2016* | GSE74912 | |
| Sequence-based reagent | sg3'HS1-3' | Synthego | Synthesized guide RNA | GAGUCUUGGGAUGGCUGAAG |
| Sequence-based reagent | sg3'HS1-5' | Synthego | Synthesized guide RNA | GUCCAAGGCAGGACAUGUGU |
| Sequence-based reagent | sgHS5-5' | Synthego | Synthesized guide RNA | GGCACCCACCUUCAAUCAAA |
| Sequence-based reagent | sgHS5-3' | Synthego | Synthesized guide RNA | AGUCCUGCCAGAUAUAGGUC |
| Sequence-based reagent | sg*OR52A1*-GATA1-5' | Synthego | Synthesized guide RNA | AUGUCUUAGUGGAUAACAGA |
| Sequence-based reagent | sg*OR52A1*-GATA1-3' | Synthego | Synthesized guide RNA | CAUAUGCUCACAGUAGGAGU |
| Sequence-based reagent | sgHPFH-enhancer-5' | Synthego | Synthesized guide RNA | GGGCAUGUAGACUGUGAUGU |
| Sequence-based reagent | sgHPFH-enhancer-3': | Synthego | Synthesized guide RNA | CAUAUGCUCACAGUAGGAGU |
| Sequence-based reagent | sgBCL11A- + 58-5': | Synthego | Synthesized guide RNA | GGACUGGCAGACCUCUCCAU |
| Sequence-based reagent | sgBCL11A- + 58-3': | Synthego | Synthesized guide RNA | CUCUUACUUAUGCACACCUG |
| Sequence-based reagent | 3'HS1-deletion-genotyping forward | Eurofins Genomics | PCR primer | TCCCTGTGTGATTACTTGCTTAC |
| Sequence-based reagent | 3'HS1-deletion-genotyping reverse | Eurofins Genomics | PCR primer | AGGTCATAACCATTCAGGTAAACT |
| Sequence-based reagent | 3'HS1-inversion-genotyping forward | Eurofins Genomics | PCR primer | TCCCTGTGTGATTACTTGCTTAC |
| Sequence-based reagent | 3'HS1-inversion-genotyping reverse | Eurofins Genomics | PCR primer | GATGAACTACTTACCACTAGGGGTC |
| Sequence-based reagent | 3'HS1-WT-genotyping forward | Eurofins Genomics | PCR primer | TCCCTGTGTGATTACTTGCTTAC |
| Sequence-based reagent | 3'HS1-WT-genotyping reverse | Eurofins Genomics | PCR primer | CTTCTGACCCCTAGTGGTGTC |
| Sequence-based reagent | HPFH enhancer-deletion-genotyping forward | Eurofins Genomics | PCR primer | ACAATGGCCATATGCTCACA |
| Sequence-based reagent | HPFH enhancer-deletion-genotyping reverse | Eurofins Genomics | PCR primer | GTCCAGGTGATTTTGCTGGT |
| Sequence-based reagent | BCL11A_58 enhancer-deletion forward | Eurofins Genomics | PCR primer | GAACAGAGACCACTACTGGCAAT |
| Sequence-based reagent | BCL11A_58 enhancer-deletion forward | Eurofins Genomics | PCR primer | CTCAGAAAAATGACAGCACCA |
| Sequence-based reagent | HBB-qPCR forward | Eurofins Genomics | PCR primer | CTGAGGAGAAGTCTGCCGTTA |
| Sequence-based reagent | HBB-qPCR reverse | Eurofins Genomics | PCR primer | AGCATCAGGAGTGGACAGAT |
| Sequence-based reagent | HBD-qPCR forward | Eurofins Genomics | PCR primer | GAGGAGAAGACTGCTGTCAATG |
| Sequence-based reagent | HBD-qPCR reverse | Eurofins Genomics | PCR primer | AGGGTAGACCACCAGTAATCTG |

| Sequence-based reagent | HBE-qPCR forward | Eurofins Genomics | PCR primer | GCAAGAAGGTGCTGACTTC |
|---|---|---|---|---|
| Sequence-based reagent | HBE-qPCR reverse | Eurofins Genomics | PCR primer | ACCATCACGTTACCCAGGAG |
| Sequence-based reagent | HBG1/2-qPCR forward | Eurofins Genomics | PCR primer | TGGATGATCTCAAGGGCAC |
| Sequence-based reagent | HBG1/2-qPCR reverse | Eurofins Genomics | PCR primer | TCAGTGGTATCTGGAGGACA |
| Sequence-based reagent | ActB-qPCR forward | Eurofins Genomics | PCR primer | CCTGGCACCCAGCACAATGAAG |
| Sequence-based reagent | ActB-qPCR reverse | Eurofins Genomics | PCR primer | AAGTCATAGTCCGCCTAGAAGC |
| Sequence-based reagent | BCL11A-qPCR forward | Eurofins Genomics | PCR primer | AACCCCAGCACTTAAGCAAA |
| Sequence-based reagent | BCL11A-qPCR reverse | Eurofins Genomics | PCR primer | GGAGGTCATGATCCCCTTCT |
| Sequence-based reagent | 3'HS1 HDR template | Gene Universal | CRISPR/Cas9 knock-in HDR template | AGACATAGAGAAAGTATATT GTGTTTAAAAGACAGCTTC TTTATAATTCTATAGAACTAA AACATTCCTATTTGCCAAGG CAGTGGAGTTTTTGCTGTT CTTAGAACATAATTACTGAA AGACACGCACACATGTCCT GCCTTGGACAAAAAATTGT ATGTCCATCCTTTAAAGGT CATTCCTTTAATGGTCTTTT CTGGACCTGACCCCTAGTG GTAAGTAGTTCATCAAACTT TCTTCCCTCCCTACTTCAGT GATGCATAAGGCAGATCTG CTTTAGTGTAAGCGAGGTC AGGCCCTCAAGAGTCTTG GGATGGCTGAAGATGTAA GAACATTCTATAAGACTTG TCCAAAGAACTGACTGTT TAATGATTCTGAATATGCT AGTTCAGAGAGAATCTAT TTACCACAAACCTGAAG |
| Software, algorithm | HiC-Pro | *Servant et al., 2015* | https://github.com/nservant/HiC-Pro | |
| Software, algorithm | Juicer | *Durand et al., 2016b* | https://github.com/theaidenlab/juicer/wiki | |
| Software, algorithm | Juicebox | *Durand et al., 2016a*; *Durand et al., 2016b* | http://aidenlab.org/juicebox/ | |
| Software, algorithm | HiNT | *Wang et al., 2020* | https://github.com/parklab/HiNT | |
| Software, algorithm | STAR | *Dobin et al., 2013* | https://github.com/alexdobin/STAR | |
| Software, algorithm | edgeR | *Robinson et al., 2010* | https://bioconductor.org/packages/edgeR/ | |
| Software, algorithm | Bowtie2 | *Langmead and Salzberg, 2012* | http://bowtie-bio.sourceforge.net/bowtie2/index.shtml | |
| Software, algorithm | BWA-MEM | *Bauer et al., 2013* | http://bio-bwa.sourceforge.net/ | |
| Software, algorithm | SAMtools | *Sankaran et al., 2009* | http://samtools.sourceforge.net/ | |
| Software, algorithm | Picard Tools | | http://broadinstitute.github.io/picard/ | |

| Software, algorithm | deepTools | *Ramírez et al., 2014* | https://deeptools.readthedocs.io/en/develop/ |
| Software, algorithm | Trim Galore | | http://www.bioinformatics.babraham.ac.uk/projects/trim_galore/ |
| Software, algorithm | Trimmomatic | *Bolger et al., 2014* | http://www.usadellab.org/cms/?page=trimmomatic |

## Tissue culture of cell lines

Our HUDEP-2 cell lines were directly obtained from the cell bank of Riken Institute, Japan, The provider validates the cell line before shipment. Additionally, the cell line has been validated by HiC sequencing on the chromosome copy number (*Figure 1—figure supplement 2C*). We also validate that all cell lines used in the study are free from mycoplasma contamination. We routinely test the mycoplasma contamination status by PCR. K562 cells were maintained in RPMI medium with 10% FBS. HUDEP clone 2 (HUDEP-2) cells were cultured as previously described (*Kurita et al., 2013*). Cells were expanded in StemSpan serum-free expansion medium supplemented with 1 μM dexamethasone (D2915, Sigma), 1 μg/mL doxycycline (D9891, Sigma), 50 ng/mL human SCF, 3 units/mL EPO, and 1% penicillin/streptomycin. HUDEP-2 cells were differentiated in a two-phase differentiation protocol consisting of IMDM supplemented with 5% human AB serum, 10 μg/mL recombinant human insulin, 330 μg/mL holo-transferrin, 3 units/mL EPO, 1 μg/mL doxycycline, 2 units/mL heparin, and 1% penicillin/streptomycin. 50 ng/mL human SCF was included in phase 1 of the culture (days 1–3) and withdrawn in phase 2 of the culture (days 4 and beyond). Samples were collected on day 5 for flow cytometry, RNA extraction, immunoblot, HiC, ATAC-seq, and HPLC analyses.

## CRISPR/Cas9-mediated deletion and homologous recombination

Cas9 nuclease (1081058, IDT) were mixed with synthetically modified sgRNAs (synthesized by Synthego) at a 1:3 molar ratio in resuspension buffer T and incubated at 37 °C for 10 min to form ribonucleoprotein complexes (RNPs). For the CTCF inversion clones, homology-directed repair (HDR) approach was used by adding 2 μg of dsDNA homology repair template (synthesized by Gene Universal) to the RNP complexes. dsDNA repair templates were designed to have 90–130 bp of homology arms distal and proximal to the PAM sequence.

For CD34+ cells, about $2.5 \times 10^5$ cells were harvested 24 hr after thawing, washed in HBSS (Gibco, 14170112), resuspended in RNP complexes, and electroporated at 1600 V and 3 pulses of 10 ms using the Neon Transfection system (Thermo Fisher Scientific). For HUDEP-2 cells, $2 \times 10^5$ were harvested, resuspended in RNP complexes, and electroporated at 1300 V and 1 pulse of 20 ms using the Neon Transfection system. 24 hr after electroporation, cells were harvested to assay for deletion or inversion. Genomic DNA was extracted using DirectPCR lysis reagent (102T , Viagen) followed by proteinase K treatment at 55 °C. PCR was performed using the EconoTaq PLUS 2X master mix (30033-2, Lucigen) with the following cycling conditions: 95 °C for 2 min; 45 cycles of 95 °C for 30 s, 55 °C for 30 s, 72 °C for 40 s; 72 °C for 5 min. Amplicons were purified using the Zymoclean Gel DNA recovery kit and Sanger sequenced.

The sequence of guide RNA and other oligo sequence used to generate guideRNA, genotyping deletion, and HDR template is listed in Key resources table.

## CD34+ cell ex vivo culture and differentiation

G-CSF mobilized human peripheral blood CD34+ HSPCs were purchased from STEMCELL Technologies. Cells were thawed on day 0 into StemSpan serum-free expansion medium (09655, STEMCELL Technologies) supplemented with 100 ng/mL Flt3L (Peprotech), 50 ng/mL human stem cell factor (SCF; 300-07, Peprotech), 100 ng/mL TPO (Peprotech), and 1% penicillin/streptomycin (Gibco). Electroporation of RNP complexes was done on days 1 and 2 of the culture. Differentiation of CD34+ CD38 into erythroid progenitors was done in four phases of erythroid differentiation medium (EDM) consist of Iscove's modified Dulbecco's medium (IMDM; Gibco) supplemented with 5% human AB serum (H6914, Sigma), 10 μg/mL recombinant human insulin (I2643, Sigma), 2 units/mL heparin

(H3393, Sigma), 3 units/mL Epogen (EPO, Amgen), 330 µg/mL holo-transferrin (T4132, Sigma), and 1% penicillin/streptomycin. EDM was further supplemented with 25 ng/mL human SCF and 1 ng/mL human IL-3 (Peprotech) in phase 1 of the culture (days 4–7). IL-3 was withdrawn and human SCF is decreased to 10 ng/mL in phase 2 of the culture (days 7–11). Human SCF is further decreased to 2 ng/mL in phase 3 of the culture (days 12–16). Human SCF was withdrawn and holo-transferrin is increased to 1 mg/mL (day 17 and beyond). Cells were collected on day 21 for flow cytometry, RNA extraction, and Giemsa stain.

## Flow cytometry analysis of fetal hemoglobin protein expression

Upon differentiation, the expression of fetal hemoglobin was analyzed by intracellular flow cytometry staining. Briefly, 50,000 cells were fixed and permeabilized in CytoFast Fix/Perm buffer set (426803, BioLegend), and incubated with FITC-conjugated anti-Human Fetal hemoglobin antibody (Miltenyl Biotec, clone # REA533) in the dark at room temperature for 15 min. In addition, phenotypic characterization of cells upon differentiation was done by cell-surface antigens staining with PE anti-human CD71 (334105, BioLegend) and APC anti-human CD235a (561775, BD Biosciences) monoclonal antibodies for 30 min at 4 °C. For CD34+ cells, FITC anti-human CD233 (130-119-780, Miltenyi Biotec) was also used to assess differentiation. Cells were analyzed using CytoFLEX S flow cytometer, and FlowJo cytometry software was used for data visualization.

## RNA isolation and quantitative PCR with reverse transcription (RT-qPCR)

RNA was extracted from 1 to 2 million cells using TRIzol (Invitrogen) followed by phenol-chloroform extraction. Reverse transcription reactions were performed with random hexamers using iScript (Bio-Rad). BCL11A, HBB, HBD, HBG1/2, and HBE mRNAs were quantified by SsoAdvanced Universal SYBR Green Supermix (1725272, Bio-Rad) and run on a CFX96 Touch Real-Time PCR Detection System (Bio-Rad). The sequence of oligos is listed in key resources table.

## Immunoblot

Protein samples were denatured in 2 X Laemmli buffer (161-0737, Bio-Rad) and boiled for 10 min. They were resolved on Novex Tris-Glycine gel (Invitrogen) and transferred onto 0.45 µM PVDF membrane (Immobilon-FL PVDF) using Invitrogen Mini Blot Module. Immunoblotting was performed with the following antibodies: BCL11A (ab191401, Abcam), beta-actin (20536-1-AP, Proteintech), β-globin (sc-21757, Santa Cruz Biotechnology), γ-globin (sc-21756, Santa Cruz Biotechnology), and ZTB7A (mab3496, R&D Systems). Starbright B700-conjugated goat α-rabbit IgG (12004161, Bio-Rad) and DyLight 800 goat α-mouse IgG (STAR117D800GA, Bio-Rad) secondary antibodies were purchased from Bio-Rad. Signals were visualized on ChemiDoc MP imaging system (Bio-Rad).

## Hemoglobin HPLC

1 million HUDEP-2 cells were harvested and washed with PBS two times. After the harvest of the cells, cells were snap frozen at –80 °. Frozen cell pellets were collected for HPLC at the University of Michigan Hospital.

## RNA-sequencing

Total RNA from HUDEP-2 cells upon 5 days of differentiation were isolated using TRIzol followed by phenol-chloroform extraction. Sequencing libraries were prepared from 500 ng of total RNA using Swift Biosciences rapid RNA library kit following the manufacturer's protocol. Libraries were sequenced in the Illumina HiSeq 4000 platform to generate paired-end reads of 2 × 150 bp.

Paired-end reads were trimmed with trim galore and aligned to hg19 genome with STAR (v2.7.0f) using the default parameters with the following modifications:(1) sjdbOverhang was set to sequence length – 1 as recommended in the STAR manual, (2) twopassMode was set to Basic, (3) outReadsUnmapped was set to None, (4) outSAMtype was set to BAM SortedByCoordinate, and (5) quantMode was set to GeneCounts. Gene expression was quantified using STAR's built-in and counts were imported into R (v4.0) using readDGE function to produce DGE list object (*Dobin et al., 2013*; *Robinson et al., 2010*).

Differential expression (DE) analysis was performed in R with the edgeR package (v3.30.3). A paired DE analysis was performed to assess changes between groups (3'HS1 deletion or inversion versus WT). Normalization factors and effective library size were applied. Dispersion was estimated using the "estimateDisp" function, with the design matrix as: ~ replicate + group, where "replicate" refers to biological replicates of each sample and "group" refers to the individual clones of deletion or inversion of 3'HS1 and WT. Likelihood ratio test was performed for differential expression with the "glmFit" and "glmLRT" functions. The list of DE genes was further filtered by setting p-value < 0.01 and absolute value of log2 fold-change >1.

## ATAC-sequencing

ATAC-seq was performed with Illumina Nextera XT library preparation kit (FC-131-1024, Illumina) as previously described (*Buenrostro et al., 2015*). Upon 5 days of HUDEP-2 differentiation, 50,000 cells were harvested and permeabilized in 50 µL of cold lysis buffer (10 mM Tris-HCl, pH 7.4, 10 mM NaCl, 3 mM MgCl₂, 0.1% IGEPAL CA-630). The transposition reaction was carried out at 37 °C for 30 min with agitation in 50 µL volume containing 25 µL of 2× TD buffer and 2.5 µL of Nextera Tn5 transposase. DNA was purified with Qiagen MinElute PCR purification kit. Library amplification was done with KAPA HiFi HotStart ReadyMix PCR kit, and the resulting libraries were purified with Qiagen MinElute PCR Purification Kit. Libraries were sequenced in the Illumina HiSeq 4000 platform to generate paired-end reads of 2 × 150 bp.

Paired-end reads were trimmed using Trim Galore (version 0.6.1) and aligned to hg19 using Burrows-Wheeler Aligner (bwa-mem, version 0.7.17). The resulting alignments were sorted, indexed using SAMtools (version 1.9), and marked for duplicates with samblaster (version 0.1.24). Reads were then normalized using deeptools bamCoverage with RPGC parameter and visualized with IGV.

## CUT&RUN

CTCF CUT&RUN was performed according to published protocol (*Meers et al., 2019*). Briefly, upon 5 days of differentiation, 500,000 HUDEP2 cells were immobilized with BioMag Plus Concanavalin A (BangsLabs, Inc). Cells were permeabilized and incubated with CTCF antibody (ab70303, Abcam) overnight. After washing away unbound antibody, pA-MNase (a gift from Dr. Steven Henikoff) was then added to the cells and incubated for 5 min in a metal block on ice. The MNase reaction was stopped, chromatin was released by diffusion at 37° C for 30 min, and DNA was extracted. Swift Biosciences Accel-NGS 2S Plus DNA library kit was used to construct NGS libraries. Libraries were sequenced on the Illumina HiSeq 4000 platform to generate paired-end reads of 2 × 150 bp.

Paired-end sequencing reads were trimmed using Trimmomatic (version 0.36) and aligned to hg19 genome assembly using Bowtie2 (version 2.3.5) with the parameter "--dovetail --phred33". The resulting alignments were indexed, sorted, and marked for duplicates with Picard "MarkDuplicates" function. Reads were then normalized using deeptools bamCoverage with RPGC parameter and visualized with IGV.

## Chromatin immunoprecipitation

About 2 million HUDEP-2 cells upon 5 days of differentiation were harvested and fixed with 1% formaldehyde for 10 min at room temperature. Fixation was quenched with 0.125 M glycine. Cells were washed twice with ice-cold PBS, lysed in 0.13 mL of lysis buffer (10 mM Tris pH 8.0, 0.25% SDS, 2 mM EDTA, 1X protease inhibitors), and sonicated in a Covaris microtube with Covaris ultrasonicator (E220, Covaris). Sonicated chromatin was diluted with 0.2 mL of equilibration buffer (10 mM Tris pH 8.0, 233 mM NaCl, 1.66% Triton X-100, 0.166% sodium deoxycholate, 1 mM EDTA, 1X protease inhibitors) and spun down to pellet insoluble materials. Supernatant was mixed with 2 µg of antibody (H3K27ac, D5E4, Cell Signaling) and incubated at 4° C overnight. 10 µL of Dynabeads protein A (Thermo Fisher Scientific) were washed twice with 0.1% BSA/PBS and incubated overnight alongside the chromatin. After overnight rotating, beads were transferred to the tube containing chromatin and incubated for 2 hr. Beads were washed twice with RIPA-LS (10 mM Tris pH 8.0, 140 mM NaCl, 1 mM EDTA, 0.1% SDS, 0.1% sodium deoxycholate, 1% Triton X-100), twice with RIPA-HS (10 mM Tris pH 8.0, 500 mM NaCl, 1 mM EDTA, 0.1% SDS, 0.1 % sodium deoxycholate, 1% Triton X-100), twice with RIPA-LiCl (10 mM Tris pH 8.0, 250 mM LiCl, 1 mM EDTA, 0.5% NP-40, 0.5% sodium deoxycholate), and once with 10 mM Tris pH 8.0. After washing, 48 µL

of elution buffer (10 mM Tris pH 8.0, 5 mM EDTA, 300 mM NaCl, 0.4% SDS) with 2 µL of 20 mg/mL of proteinase K (Viagen) were added to the beads and incubated for 1 hr at 55 °C followed by overnight incubation at 65 °C to decrosslink. Beads were magnetized and supernatant was purified with phenol-chloroform. DNA was precipitated with ice-cold absolute ethanol, washed with 75% ethanol, and eluted with 20 µL of 10 mM of Tris pH 8.0. Accel-NGS 2S Plus DNA library kit (21096, Swift Bioscience) was used to construct the libraries with the 2S combinatorial dual indexing kit (28096, Swift Bioscience). Libraries were sequenced on the HiSeq 4000 platform to generate paired-end reads sequencing 2 × 150 bp.

Reads were aligned to the human genome (hg19) using Bowtie2 (version 2.3.5). The resulting alignments bam files were indexed, sorted with SAMtools, and marked for duplicates with Picard "MarkDuplicates" function. Reads were then normalized using deeptools bamCoverage with RPGC parameter and visualized with IGV.

## Chromatin conformation capture (HiC and capture HiC)

### Hi-C library preparation

Approximately 2 million cells of each HUDEP-2 clone were differentiated for 5 days and fixed in 1% formaldehyde. Hi-C libraries were generated using the Arima-HiC kit, according to the manufacturer's protocols. Libraries were prepared using the Accel-NGS 2S Plus kit (Swift Biosciences, 21096), with single indexing kit 2S Set A (Swift Biosciences, 26148). The final amplification cycle numbers for each library were determined by qPCR in the QC2 step of the Arima protocol. Quantification of the libraries was performed with KAPA Library Quantification Kit (Roche, KK4824). The libraries were then pooled for sequencing on NovaSeq S4 to get between 300 and 400 million reads each.

### Probe design

The capture probes were designed as previously described (*Sanborn et al., 2015*). 3822 oligos of probe sequence covering Chr11:4665299–5954156 with adaptor sequence on both end as follows: ATCGCACCAGCGTGT N120 CACTGCGGCTCCTCA was synthesized by GeneScripts. The biotinylated RNA probes specific to the β-globin locus were made as described. Briefly, the desired sequences were amplified out of the pool with primer sequences complementary to both ends using KAPA HiFi HotStart MasterMix for 12 cycles. The oligonucleotides were then prepared for in vitro transcription by adding a T7 promoter to the forward primer and amplifying for a further 15 cycles. In vitro transcription with Biotin-16-UTP was performed with NEB HiScribe T7 for 2 hr at 37 °C. The template DNA was degraded by adding 1 µL DNase and incubating at 37 °C for a further 15 min, and then stopping the reaction by adding 1 µL 0.5 M EDTA. The RNA was then purified using Zymo RNA Clean & Concentrator columns, eluting in 15 µL elution buffer. We then added 1 U/µL RNase Inhibitor (NEB, M0314), aliquoted, and stored at –80 °C until needed for the capture.

### Capture

150–500 ng of the previously created Hi-C libraries were diluted to 25 µL and mixed with 2.5 µg Cot-1 DNA and 10 µg salmon sperm DNA and heated to 95 °C for 5 min, then held at 65 °C for at least 5 min. 33 µL of prewarmed (to 65 °C) hybridization buffer (10× SSPE, 10× Denhardt's buffer, 10 mM EDTA, and 0.2% SDS), along with 6 µL of RNA probe mixture (500 ng biotinylated-RNA probes and 20 U RNase inhibitor) were added to the DNA mixture and hybridized for 24 hr at 65 °C. After the hybridization incubation was complete, 50 µL of Streptavidin T1 beads (Dynabeads, Life Technologies) were washed in Bind-and-Wash buffer (1 M NaCl, 10 mM Tris-HCl, pH 7.5, and 1 mM EDTA), resuspended in 134 µL of the same buffer, and then added to the hybridization mixture. The beads were allowed to bind to the biotinylated, hybridized DNA and RNA mixture for 30 min at room temperature before separating and discarding the supernatant. The bead-bound DNA was then washed once with low-stringency wash buffer (1× SSC, 0.1% SDS), for 15 min at room temperature, and three times with high-stringency wash (0.1× SSC, 0.1% SDS) for 10 min at 65 °C, separating the beads on a magnet each time before discarding the supernatant. After the last wash, the beads were resuspended in 21 µL nuclease-free water. 1 µL was diluted 1:1000 and had qPCR performed as for QC2 of the Arima protocol to determine the number of cycles to amplify the enriched libraries.

Virtual 4C

Virtual 4C track was generated by using Juicebox. Horizontal and vertical 1D track of the 5 kb (chr11:5,165,001–5,170,000) × 5 kb (chr11:5,275,001–5,280,000) pixel was generated with Juicebox 'generate 1D track' function.

## Acknowledgements

We thank Dr. James Engel for valuable discussions and technical expertise on the HUDEP-2 cell culture and differentiation. XZ was supported by the VARI fellowship. XZ was supported by the ASH scholar award and is an EvansMDS Young Investigator.

## Additional information

### Funding

| Funder | Grant reference number | Author |
|---|---|---|
| National Human Genome Research Institute | 1R35HG011279-01 | Jie Liu |
| Van Andel Research Institute | Fellow program | Xiaotian Zhang |
| American Society of Hematology | Scholar Program | Xiaotian Zhang |

The funders had no role in study design, data collection and interpretation, or the decision to submit the work for publication.

### Author contributions

Pamela Himadewi, Data curation, Formal analysis, Investigation, Methodology, Validation, Visualization, Writing – original draft; Xue Qing David Wang, Investigation, Writing – original draft; Fan Feng, Data curation, Formal analysis, Software, Writing – original draft, Writing - review and editing; Haley Gore, Methodology, Writing – original draft; Yushuai Liu, Investigation; Lei Yu, Methodology; Ryo Kurita, providing material; Yukio Nakamura, provide research material; Gerd P Pfeifer, Jie Liu, Supervision, Writing - review and editing; Xiaotian Zhang, Conceptualization, Data curation, Formal analysis, Funding acquisition, Investigation, Methodology, Project administration, Supervision

### Author ORCIDs

Gerd P Pfeifer http://orcid.org/0000-0002-5080-9604
Jie Liu http://orcid.org/0000-0002-9504-0587
Xiaotian Zhang http://orcid.org/0000-0002-9533-4761

### Decision letter and Author response

Decision letter https://doi.org/10.7554/eLife.70557.sa1
Author response https://doi.org/10.7554/eLife.70557.sa2

## Additional files

### Supplementary files

• Transparent reporting form

• Source data 1. Blot and Gel images.

• Supplementary file 1. Differential expressed genes in 3'HS1 deletion and inversion clones. Differential expressed genes ( |log2Fold change| >1 and p-value <0.01) in 3'HS1 deletion and inversion clones.

### Data availability

Sequencing data have been deposited in GEO under accession codes GSE160425.

The following dataset was generated:

| Author(s) | Year | Dataset title | Dataset URL | Database and Identifier |
|---|---|---|---|---|
| Zhang X, Himadewi P, Gore H | 2020 | Chromosomal loop engineering in human beta globin locus | https://www.ncbi.nlm.nih.gov/geo/query/acc.cgi?acc=GSE160425 | NCBI Gene Expression Omnibus, GSE160425 |

The following previously published datasets were used:

| Author(s) | Year | Dataset title | Dataset URL | Database and Identifier |
|---|---|---|---|---|
| Zhu Q, Liu N, Hargreaves V, Orkin S | 2017 | Direct Promoter Repression by BCL11A Controls the Fetal to Adult Hemoglobin Switch | https://www.ncbi.nlm.nih.gov/geo/query/acc.cgi?acc=GSE104676 | NCBI Gene Expression Omnibus, GSE104676 |
| Martyn GE, Wienert B, Yang L, Shah M, Norton LJ, Burdach J, Kurita R, Nakamura Y, Pearson RC, Funnell AP, Quinlan KG, Crossley M | 2017 | Natural regulatory mutations elevate fetal globin via disruption of BCL11A or ZBTB7A binding | https://www.ncbi.nlm.nih.gov/geo/query/acc.cgi?acc=GSE103445 | NCBI Gene Expression Omnibus, GSE103445 |
| Buenrostro J | 2016 | ATAC-seq data | https://www.ncbi.nlm.nih.gov/geo/query/acc.cgi?acc=GSE74912 | NCBI Gene Expression Omnibus, GSE74912 |

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
