## [Decision Letter]

**Acceptance summary:**

This study provides insight into specifics of the organization and regulation of the human β-globin gene locus, and in particular of the contribution of a CTCF binding site located immediately downstream of the cluster. In addition to clarifying and expanding upon prior studies related to deletional hereditary persistence of fetal hemoglobin (HPFH), this work provides an interesting example of changes to gene expression and locus-specific chromatin organization upon manipulation of a CTCF binding site.

**Decision letter after peer review:**

Thank you for submitting your article "3'HS1 CTCF binding site in human β-globin locus regulates fetal hemoglobin expression" for consideration by *eLife*. Your article has been reviewed by 2 peer reviewers, one of whom is a member of our Board of Reviewing Editors, and the evaluation has been overseen by Jessica Tyler as the Senior Editor. The reviewers have opted to remain anonymous.

Essential revisions:

1) Present expression data for the genes within the β-globin cluster in different backgrounds as requested by the reviewers, including HBB expression in the 48 kb deletion and in the 3'HS1 deletion performed in the primary cell background.

2) Address the concerns of both reviewers concerning the clarity of the presentation of the Hi-C data and the interpretation of the ATAC- and H3K27ac ChIP-seqs, as well as the relative effects of the enhancer deletion on different genes.

3) Address or rebut the concern of reviewer #1 regarding the number of DEGs observed upon deletion of 3'HS1.

*Reviewer #1 (Recommendations for the authors):*

1. (Lines 86-87, Figure 1) It is somewhat confusing for the authors to jump into Hi-C results immediately after introducing the derivation of single cell-derived HUDEP-2 subclones. Having shown expression of β-globin genes in CRISPR/Cas9-manipulated HUDEP-2 bulk populations (which is of limited utility here and might productively be omitted), one expects similar analysis of the subclones first. Instead, this is practically an afterthought in Figures G-H. The presentation in these panels is also odd, in that 2 3'HS1-deleted subclones are presented as relative HBB/HBG expression in G, and then a different one is presented as normalized expression of HBE/G/B in H. Why not show all 3 subclones in a clearer fashion? Also, while normalized expression certainly gives an idea of the magnitude of changes upon 3'HS1 deletion, given the authors' speculations regarding potential therapeutic ramifications of increased HBG expression here, it would be worth at least a supplemental panel showing the absolute levels of expression of each of these genes.

2. (Lines 124-129) Although it seems likely that the authors' statement is accurate thatincreased accessibility, as indicated by ATAC-seq, and increased H3K27Ac, occur over the bodies of the HBG genes upon 3'HS1 deletion (Figure S4A), the figure appears to show increases in accessibility at the HBB/D genes as well, and increases in H3K27Ac peaks both at the genes and across the LCR.

3. (Lines 128-129) "…and the region with significant peak increase is not associated with BCL11A 129 binding at the HBG1/2 promoter region." This is confusingly worded. What the authors mean here is that the regions over the HBG genes associated with increased reads in the ATAC-seq and H3K27Ac ChIP-seqs do not map to the regions associated with BCL11A binding. In addition, however, this is an odd observation, which the authors do not attempt to clarify – that is, how is HBG expression increasing without any noticeable changes in accessibility at the promoter region? Is it just that the absolute levels of HBG expression are so low, and/or occurring in such a small proportion of cells, that such changes aren't being detected?

4. (Lines 133-135) Is it surprising to find so many DEGs upon deletion of 3'HS1? It seems odd to observe more than 300 DEGs after deletion of a single CTCF binding site within the β-globin locus. None of the nearby genes are known transcriptional regulators that might be expected to exert significant downstream effects on multiple additional genes, and based on the current literature one does not expect loss of a single CTCF binding site to exert such widespread effects beyond its specific location in the genome. What are these DEGs – is there any significance that emerges upon pathway analysis? Where are they located in the genome?

5. (Line 146) There is no Figure 2G; the authors are likely referring to Figure 2F here.

6. (Line 151) I'm not sure that "synergizes" is an accurate term here. The results suggest that 3'HS1 deletion results in upregulation of HBG expression via a mechanism unrelated to BCL11A-mediated transcriptional repression, though.

7. (Lines 173-180, Figures 3C-D) The GATA-1/enhancer deletions appear to affect HBE and HBB to the same or nearly the same degree as HBG.

8. (Lines 199-201) What is the effect of the 48 kb deletion on HBB?

9. (Line 227) This is likely not a proper TAD – TADs are generally much larger than the regions the authors describe here. "Sub-TAD" (as per terminology commonly used by the Higgs group with relation to locus organization at the α-globin locus) might be more appropriate.

10. (Lines 236-239) Where might the additional enhancers the authors refer to here actually be? There is nothing in Figure 3A that might suggest the presence of such enhancers. Can the authors clarify this further?

*Reviewer #2 (Recommendations for the authors):*

The authors analyzed the functional role of a CTCF binding site in the β-globin gene locus in Hudep-2 and differentiating CD34+ cells. Previous studies have shown that CTCF sites flanking the globin gene locus interact and form chromosomal loops. The authors found that deleting 3'HS1 specifically increased expression of the γ-globin gene and reduced expression of the β-globin gene. This seems independent from the levels of the known γ-globin repressor BCL11a. Through analysis of ATAC seq. and GATA1-ChIP-seq. data, they identified an enhancer that upon deletion reduced activation of globin in the 3'HS1 deficient cells. Deletion of a GATA1 site within this enhancer also reduced globin expression.

This is an interesting manuscript that provides functional insight into regulatory DNA elements located downstream of the β-globin gene cluster. This study is also significant with respect to potentially improving therapeutic fetal globin production. Overall, the experiments include appropriate controls and statistics.

The following issues should be address:

1. It is somewhat difficult to follow the authors interpretation of the Hi-C data in Figure 1 C-E. It appears that upon deletion of 3'HS1 a domain is formed between the most 3' CTCF site (OR52A5-CBS) and a sequence upstream of 3'HS1. This domain appears to contain the adult β-globin gene and there are more frequent interactions within this domain in 3'HS1 deleted cells, compared to WT cells. Could this in part explain why this deletion increases γ-globin and not β-globin. Likewise, the green-dotted circle seems to highlight interactions of OR52A5-CBS with a region upstream of HS5, but not with HS5.

2. The authors show that deletion or inversion of 3'HS1 does not affect expression of BCL11A. However, the western blot data in Figure 2E suggest that there is more BCL11A in the 3'HS1 inverted clones. This may explain why there is less γ-globin expressed in these cells.

3. Does deletion of 3'HS1 affect expression of β-globin in differentiating CD34+ cells.

---

## [Author Response]

Essential revisions:1) Present expression data for the genes within the β-globin cluster in different backgrounds as requested by the reviewers, including HBB expression in the 48 kb deletion and in the 3'HS1 deletion performed in the primary cell background.

We have now added these data in the Figure 1H, Supplemental Figure 6 and Figure 4.

2) Address the concerns of both reviewers concerning the clarity of the presentation of the Hi-C data and the interpretation of the ATAC- and H3K27ac ChIP-seqs, as well as the relative effects of the enhancer deletion on different genes.

We have updated the HiC presentation and clarify the interpretation of ATAC-seq and ChIP-seq data.

3) Address or rebut the concern of reviewer #1 regarding the number of DEGs observed upon deletion of 3'HS1.

We have responded the concern of reviewer #1 with GO term analysis of DEGs upon the deletion of 3’HS1.

Reviewer #1 (Recommendations for the authors):1. (Lines 86-87, Figure 1) It is somewhat confusing for the authors to jump into Hi-C results immediately after introducing the derivation of single cell-derived HUDEP-2 subclones. Having shown expression of β-globin genes in CRISPR/Cas9-manipulated HUDEP-2 bulk populations (which is of limited utility here and might productively be omitted), one expects similar analysis of the subclones first. Instead, this is practically an afterthought in Figures G-H. The presentation in these panels is also odd, in that 2 3'HS1-deleted subclones are presented as relative HBB/HBG expression in G, and then a different one is presented as normalized expression of HBE/G/B in H. Why not show all 3 subclones in a clearer fashion? Also, while normalized expression certainly gives an idea of the magnitude of changes upon 3'HS1 deletion, given the authors' speculations regarding potential therapeutic ramifications of increased HBG expression here, it would be worth at least a supplemental panel showing the absolute levels of expression of each of these genes.

We appreciate the reviewer’s suggestion on the reorganization of the panels in Figure 1. We wish to first emphasize how the alteration of CTCF leads to changes in 3D genomic

structure. We have now presented the absolute expression data of HBB/HBG/HBE from individual clones in Figure 1H.

2. (Lines 124-129) Although it seems likely that the authors' statement is accurate thatincreased accessibility, as indicated by ATAC-seq, and increased H3K27Ac, occur over the bodies of the HBG genes upon 3'HS1 deletion (Figure S4A), the figure appears to show increases in accessibility at the HBB/D genes as well, and increases in H3K27Ac peaks both at the genes and across the LCR.

We appreciate the reviewer’s comment on the ATAC and H3K27ac level in the 3’HS1 deletion and inversion clones. In the H3K27ac comparison, the signal at the HBB locus in WT is saturated and thus the signal is actually higher in WT than 3’HS1 deletion. So, with some variability, we believe that these subtle changes in ATAC and H3K27ac data is likely due to the clonal variation seen in HUDEP-2 cells before, and it is not necessarily an indicative of the functional gene expression alteration.

3. (Lines 128-129) "…and the region with significant peak increase is not associated with BCL11A 129 binding at the HBG1/2 promoter region." This is confusingly worded. What the authors mean here is that the regions over the HBG genes associated with increased reads in the ATAC-seq and H3K27Ac ChIP-seqs do not map to the regions associated with BCL11A binding. In addition, however, this is an odd observation, which the authors do not attempt to clarify – that is, how is HBG expression increasing without any noticeable changes in accessibility at the promoter region? Is it just that the absolute levels of HBG expression are so low, and/or occurring in such a small proportion of cells, that such changes aren't being detected?

We apologize for the confusion caused by the sentence. We have rephrased this sentence “There was also a significant increase in activating H3K27ac in the HBG2 gene body, and the region with significant peak increase is not associated with BCL11A binding at the HBG1/2 promoter region” into “There was also a significant increase in activating H3K27ac in the HBG2 gene body. Interestingly, these epigenetic changes upon deletion of 3’HS1 CBS does not occur at the promoter of HBG2 which suggests that this CTCF-derived regulation is independent from BCL11A transcriptional regulation.” to clarify the statement.

4. (Lines 133-135) Is it surprising to find so many DEGs upon deletion of 3'HS1? It seems odd to observe more than 300 DEGs after deletion of a single CTCF binding site within the β-globin locus. None of the nearby genes are known transcriptional regulators that might be expected to exert significant downstream effects on multiple additional genes, and based on the current literature one does not expect loss of a single CTCF binding site to exert such widespread effects beyond its specific location in the genome. What are these DEGs – is there any significance that emerges upon pathway analysis? Where are they located in the genome?

We appreciate the reviewer’s constructive comments. We performed gene ontology analysis on these genes and no significant enriched pathway (FDR <0.1). The top enriched pathway in upregulated genes is associated with hemoglobin genes (shown in Author response image 1). These genes are randomly distributed on the genome without enrichment in particular chromosome (shown in Author response image 1). In summary, we believe that the differential expressed genes may be associated with clone-to-clone variation.

**Author response image 1. sa2fig1:** 

5. (Line 146) There is no Figure 2G; the authors are likely referring to Figure 2F here.

We apologize for the mistake. It is referring to Figure 2F here. This has been fixed.

6. (Line 151) I'm not sure that "synergizes" is an accurate term here. The results suggest that 3'HS1 deletion results in upregulation of HBG expression via a mechanism unrelated to BCL11A-mediated transcriptional repression, though.

We have rephrased the word “synergizes” with “accentuates.”

7. (Lines 173-180, Figures 3C-D) The GATA-1/enhancer deletions appear to affect HBE and HBB to the same or nearly the same degree as HBG.

We appreciate the reviewer’s comment on this data. Indeed, in comparison with WT, we observed 60% decrease in HBB expression with HPFH enhancer deletion in comparison to Δ3’HS1 cells. We believe that upon the deletion of 3’HS1, HPFH enhancer is incorporated into the regulation hub of β-globin cluster. As a result, its activity might have contributed to the HBB expression. We also believe that further dissection of HPFH enhancer regulation will require systematic investigation using high resolution 3D genomic technology like Micro-C, which is outside the scope of this paper.

8. (Lines 199-201) What is the effect of the 48 kb deletion on HBB?

It appears that the 48 kb deletion results in the reduction of HBB expression to about 50% in comparison to WT. This may be due to the compensation effect of HBG and HBE upregulation. We have now added the HBB expression in the Supplemental Figure S6 panel C.

9. (Line 227) This is likely not a proper TAD – TADs are generally much larger than the regions the authors describe here. "Sub-TAD" (as per terminology commonly used by the Higgs group with relation to locus organization at the α-globin locus) might be more appropriate.

We appreciate the reviewer’s comment, as a relatively new concept, TAD and sub-TAD definitions are still debated on many occasions. We have changed the TAD to sub-TAD and cite the reference accordingly as suggested by reviewer #1.

10. (Lines 236-239) Where might the additional enhancers the authors refer to here actually be? There is nothing in Figure 3A that might suggest the presence of such enhancers. Can the authors clarify this further?

We apologize for the confusion. Based on the GATA1 binding and ATAC-seq data in mature erythroid cells, we observed GATA1 binding site which co-occurs together with ATAC-seq peaks (Author response image 2). The active chromatin landscape in the regions between OR52A1 and OR52A5 CTCFs suggests that these sites may acts as potential cis regulator elements (CRE) for globin gene expression. Yet we have not validated the function of these potential CREs as bona fide enhancers through comprehensive gene editing.

Reviewer #2 (Recommendations for the authors):The authors analyzed the functional role of a CTCF binding site in the β-globin gene locus in Hudep-2 and differentiating CD34+ cells. Previous studies have shown that CTCF sites flanking the globin gene locus interact and form chromosomal loops. The authors found that deleting 3'HS1 specifically increased expression of the γ-globin gene and reduced expression of the β-globin gene. This seems independent from the levels of the known γ-globin repressor BCL11a. Through analysis of ATAC seq. and GATA1-ChIP-seq. data, they identified an enhancer that upon deletion reduced activation of globin in the 3'HS1 deficient cells. Deletion of a GATA1 site within this enhancer also reduced globin expression.This is an interesting manuscript that provides functional insight into regulatory DNA elements located downstream of the β-globin gene cluster. This study is also significant with respect to potentially improving therapeutic fetal globin production. Overall, the experiments include appropriate controls and statistics.

We appreciate the reviewer for the comments on our manuscript.

The following issues should be address:1. It is somewhat difficult to follow the authors interpretation of the Hi-C data in Figure 1 C-E. It appears that upon deletion of 3'HS1 a domain is formed between the most 3' CTCF site (OR52A5-CBS) and a sequence upstream of 3'HS1. This domain appears to contain the adult β-globin gene and there are more frequent interactions within this domain in 3'HS1 deleted cells, compared to WT cells. Could this in part explain why this deletion increases γ-globin and not β-globin. Likewise, the green-dotted circle seems to highlight interactions of OR52A5-CBS with a region upstream of HS5, but not with HS5.

We apologize for the lack of clarity in our interpretation of the data. We have now realigned the regions in Figure 1C-E.

2. The authors show that deletion or inversion of 3'HS1 does not affect expression of BCL11A. However, the western blot data in Figure 2E suggest that there is more BCL11A in the 3'HS1 inverted clones. This may explain why there is less γ-globin expressed in these cells.

We appreciate the reviewer’s comments on BCL11A level in the western blot. We performed density quantification on the western blot bands. The 3’HS1 inversion clone has slightly higher BCL11A protein level with 10-20% increase in density. We believe that this slight increase of BCL11A protein level would not have a substantial impact on the globin gene expression (Author response image 3) .

**Author response image 3. sa2fig3:** 

3. Does deletion of 3'HS1 affect expression of β-globin in differentiating CD34+ cells.

The effect of the 3’HS1 deletion on β-globin is heterogenous in CD34+ cells between individual donors. We observed both increase and decrease of β-globin expression across different samples as shown in (Author response image 4) .

**Author response image 4. sa2fig4:**